# SERPINA3: A Novel Therapeutic Target for Diabetes-Related Cognitive Impairment Identified Through Integrated Machine Learning and Molecular Docking Analysis

**DOI:** 10.3390/ijms26051947

**Published:** 2025-02-24

**Authors:** Yu An, Zhaoming Cao, Yage Du, Guangyi Xu, Jingya Wang, Yinchao Ma, Ziyuan Wang, Jie Zheng, Yanhui Lu

**Affiliations:** 1Department of Endocrinology, Beijing Chao-Yang Hospital, Capital Medical University, Beijing 100020, China; anyu900222@126.com; 2School of Nursing, Peking University, Beijing 100191, China; 2311110250@bjmu.edu.cn (Z.C.); xuguangyi1998@163.com (G.X.); 2411210127@stu.pku.edu.cn (J.W.);; 3NHC Key Laboratory of Medical Immunology, School of Basic Medical Sciences, Peking University, Beijing 100191, Chinalegolovers@pku.edu.cn (Z.W.); 4Beijing Life Science Academy, Beijing 100191, China

**Keywords:** diabetes, cognitive impairment, SERPINA3, machine learning, sulfonylurea drugs, molecular docking, neuroprotection

## Abstract

Diabetes-related cognitive impairment (DCI) is a severe complication of type 2 diabetes mellitus (T2DM), with limited understanding of its molecular mechanisms hindering effective therapeutic development. This study identified SERPINA3 as a potential therapeutic target for DCI through integrated machine learning and molecular docking analyses. Transcriptomic data from cortical neuronal samples of T2DM patients were analysed using support vector machine recursive feature elimination (SVM-RFE) and least absolute shrinkage and selection operator (LASSO) regression, revealing SERPINA3 as a significantly upregulated gene in DCI. Experimental validation via Western blot confirmed elevated SERPINA3 protein levels in DCI patient plasma. Molecular docking demonstrated the stable binding of sulfonylurea hypoglycaemic agents, such as gliclazide and glimepiride, to SERPINA3, with binding energies of −6.8 and −6.6 kcal/mol, respectively. These findings suggest that SERPINA3 plays a pivotal role in DCI pathogenesis and that sulfonylurea drugs may exert neuroprotective effects through SERPINA3-mediated pathways. This study provides novel insights into the molecular mechanisms of DCI and highlights the potential of SERPINA3-targeted therapies for early intervention and treatment. Further research is warranted to validate these findings in larger cohorts and explore their clinical applicability.

## 1. Introduction

Diabetes-related cognitive impairment (DCI), one of the most serious complications of type 2 diabetes mellitus (T2DM), has become a major challenge in global public health. Epidemiological studies have shown that the risk of cognitive impairment in T2DM patients is 1.5–2.5 times greater than that in nondiabetic individuals [1]. Moreover, the incidence of DCI is increasing as the global population is aging. According to the International Diabetes Federation (IDF), it is predicted that by 2045, the number of diabetic patients worldwide will reach 783 million, of which approximately 30–40% may develop DCI [2].

The pathogenesis of DCI is complex and involves multiple molecular pathways and signalling cascade responses. Recent studies have shown that multiple pathological processes, such as insulin resistance, the inflammatory response, oxidative stress and vascular dysfunction, are involved in the development of DCI [3]. In particular, the dysfunction of the neurovascular unit (NVU) is thought to be a key link between T2DM and cognitive dysfunction. However, current research on early diagnostic markers and therapeutic targets for DCI still has major limitations, which seriously restrict early intervention in this disease and improvements in therapeutic efficacy.

With the rapid development of bioinformatics technology, machine learning methods have shown unique advantages in the study of complex disease mechanisms. Support vector machine recursive feature elimination (SVM-RFE) has become an important tool for identifying key genes associated with diseases because of its excellent performance in high-dimensional data analysis and feature selection. Recent studies have shown that SVM-RFE-based biomarker screening models exhibit high accuracy and reliability in a variety of complex diseases [4]. However, relatively few studies have applied SVM-RFE to DCI key gene identification, and a systematic experimental validation and exploration of the underlying molecular mechanism is lacking.

In this study, we applied the SVM-RFE method combined with the least absolute shrinkage and selection operator (LASSO) regression algorithm for the first time to systematically screen DCI-related key genes on the basis of transcriptomic data from cortical neuronal samples of T2DM patients. Through a combination of bioinformatics analysis and experimental validation, we successfully identified and validated SERPINA3 as a potential therapeutic target for DCI. SERPINA3, a multifunctional serine protease inhibitor, plays an important role in neurological development and functional maintenance. Recent studies have shown that aberrant SERPINA3 expression is closely associated with a variety of neurodegenerative diseases [2]. However, the specific role of SERPINA3 in the pathogenesis of DCI and its potential as a therapeutic target have not been fully investigated; furthermore, through molecular docking and structural analyses, we predicted small molecule compounds that may interact with SERPINA3, which provides new ideas for the development of targeted therapeutic agents. This study not only contributes to the understanding of the molecular mechanism of DCI but also provides an important theoretical basis for the early prevention of this disease and optimisation of therapeutic strategies.

## 2. Results

### 2.1. Differentially Expressed Genes (DEGs)

A total of 106 differentially expressed genes were screened by the differential gene analysis of the GSE161355 dataset, of which 81 were upregulated genes and 25 were downregulated genes. A volcano plot revealed that the expression of up- and downregulated genes differed significantly between diabetes-related cognitive impairment (DCI) patients and normal controls (Figure 1A). The heatmap further demonstrated the expression patterns of DEGs between samples, with a clear grouping feature of gene expression between the DCI group and normal controls (Figure 1B).

### 2.2. Functional ENRICHMENT Analysis

#### 2.2.1. GO Enrichment Analysis

The GO enrichment analysis of the 106 DEGs revealed that the biological process (BP) terms of the DEGs were enriched mainly in the regulation of angiogenesis and regulation of vasculature development, and the cellular component (CC) terms were enriched mainly in structures such as the secretory granule lumen, cytoplasmic vesicle lumen and vesicle lumen. The enriched molecular function (MF) terms were mainly enzyme inhibitor activity, unfolded protein binding, and structural constituent of ribosome (Figure 2A).

#### 2.2.2. Kyoto Encyclopedia of Genes and Genomes (KEGG) Analysis

KEGG pathway enrichment analysis revealed a total of 11 significant pathways, including coronavirus disease—COVID-19, ribosome, p53 signalling pathway, thyroid cancer, FoxO signalling pathway, legionellosis, endometrial cancer, mineral absorption, longevity-regulating pathway—multiple species, basal cell carcinoma, and neuroactive ligand–receptor interaction (Figure 2B). These pathways are closely related to the pathological mechanisms of DCI, suggesting that T2DM may affect neuronal function through mechanisms such as the regulation of the cell cycle and DNA damage repair.

### 2.3. Machine Learning Screening of DCI Diagnostic Genes and Validation

The 106 differentially expressed genes were screened for DCI diagnostic genes via the LASSO regression method combined with the SVM-RFE algorithm, and a total of three DCI diagnostic genes, SERPINA3, INTU, and OXNAD1, were obtained via the intersection set (Figure 3A–C). The expression levels of the SERPINA3, INTU, and OXNAD1 genes in the training set were significantly different between the two groups (*p* values were less than 0.01, Figure 3D–F). The external data validation of these three genes revealed that the expression of SERPINA3 in the temporal lobe neurons of the two groups in the validation set was significantly different (*p* = 0.013, Figure 3G), and the results of plotting ROC curves revealed that the area under the curve of SERPINA3 was 0.808, with a 95% CI = 0.603~0.959 (Figure 3H), which suggested that the SERPINA3 has good diagnostic value for diabetes-induced degenerative changes in cognitive function.

### 2.4. WB Experiment to Verify the Expression of SERPINA3

The results of the WB experiments revealed that the expression of SERPINA3 in the plasma of DCI patients was significantly greater than that in the healthy control group (*p* < 0.01) (Figure 4). ImageJ software analysis revealed that the optical density of the SERPINA3 protein bands was significantly greater in the DCI group than in the control group, which further verified its high expression characteristics in DCI.

### 2.5. Spatial and Chemical Characterisation of the SERPINA3 Binding Site and Molecular Docking Results

The 3D structure of alpha-1-antichymotrypsin was analysed, and the RCL region (358–374) was found to be the major drug binding site. The geometric characterisation of the binding pocket revealed a volume of approximately 850 Å^3^, a depth of 12.5 Å, and an opening diameter of approximately 8.3 Å. Key residues include a hydrophobic core (Met358, Pro357, and Val373) and polar residues (Ser359 and Glu363), opening up possibilities for potential small-molecule drug binding.

Molecular docking analyses were performed to assess the affinity of clinically used hypoglycaemic drugs (metformin, gliclazide, glimepiride, repaglinide, and liraglutide) for their targets. The binding poses and interactions of the five drug candidates with the protein were obtained via AutoDock Vina v.1.2.2, and the binding energies for each interaction were generated. The results showed that the sulfonylurea hypoglycaemic agents gliclazide (Figure 5) and glimepiride (Figure 6) bind to their protein targets through visible hydrogen bonds. For alpha-1-antichymotrypsin, gliclazide and glimepiride have low binding energies of −6.6 and −6.8 kcal/mol, respectively, suggesting highly stable binding.

## 3. Discussion

In this study, we combined LASSO regression and SVM-RFE, and provided a robust framework for feature selection despite the relatively small sample size. The use of two independent methods helped mitigate potential bias and increased the reliability of our findings.

We investigated the expression of SERPINA3 in DCI and its potential mechanism of action on the basis of transcriptomics data and in vitro experimental validation and found that sulphonamides were able to bind to DCI stably via molecular structure analysis and molecular docking. This finding is consistent with previous findings reported in the literature that SERPINA3/Serpina3c has a broad regulatory function in a variety of nontumor diseases [5]. Current studies on SERPINA3/Serpina3c in the context of carcinogenesis are relatively well established and have demonstrated its involvement in multiple cellular processes, such as gene expression, the cell cycle, apoptosis, and signalling [6,7,8,9]. However, increasing evidence suggests that SERPINA3/Serpina3c also plays an equally important role in non-tumour contexts, and the mechanism of action, especially in neurodegenerative diseases [3], cardiovascular diseases [10], and metabolic disorders [11], deserves to be investigated in depth. SERPINA3, which is commonly highly expressed in the brain tissue of Alzheimer’s disease patients, is essentially a serine protease inhibitor that attenuates neuroinflammation and cellular damage by inhibiting the activity of histone G as well as other inflammation-related enzymes [12]. In the present study, we observed that the expression of SERPINA3 was significantly increased in the DCI group, suggesting that SERPINA3 may have some protective effects on coping with neuroinflammation and improving the elasticity of neural networks. Notably, the high expression of SERPINA3 may also be associated with excessive self-defence or the compensation of pathological processes. Although some studies suggest that the upregulation of SERPINA3 in neurodegenerative diseases or diabetes-associated brain injury is protective, we cannot rule out the possibility that SERPINA3 may accelerate the repair of abnormalities or induce other pathological chain reactions at certain stages of the disease. In the future, we may consider combining knockdown or overexpression techniques in dynamic intervention studies using animal models to clarify the effects of SERPINA3 on neurodegenerative processes at different stages (early, progressive, and terminal).

The development of diabetes mellitus and its complications is closely related to imbalances in blood glucose, lipids, and multiple metabolic pathways in vivo, and its core pathological processes are often accompanied by chronic inflammation and insulin resistance. SERPINA3 is associated with a variety of metabolic disorders (e.g., non-alcoholic fatty liver disease, obesity, and atherosclerosis) through the regulation of inflammatory factor release and endothelial dysfunction. Studies in the literature on the expression pattern of SERPINA3/Serpina3c in liver and adipose tissue have indicated that this gene may affect lipid metabolism and glucose homeostasis through multiple signalling axes, such as the Wnt, JNK/Nur77, and FOXO1 axes, and may play a key role in the regulation of inflammation and apoptosis [13,14,15,16]. The present study confirmed the upregulation of SERPINA3 in DCI samples, which corresponds to the combined state of elevated systemic inflammatory levels, increased insulin resistance, and impaired cerebral microvascular function in patients with type 2 diabetes mellitus. In other words, the altered expression of SERPINA3 may be an important coping mechanism when an organism faces inflammation and metabolic stress. On the basis of these findings, we hypothesised that inhibiting or regulating SERPINA3 could be a potential starting point for intervention in diabetes-related complications and delaying the process of DCI. Specifically, the body’s adaptive and reparative capacity to high glucose, high lipids and inflammation could be strengthened by targeting and regulating the interactions between SERPINA3 and its upstream/downstream signalling molecules. To verify this hypothesis, further investigations in more clinical samples and animal experiments, such as observing the dynamic relationship between the degree of abnormalities in glucose and lipid metabolism and SERPINA3 expression and confirming the bidirectional effects of regulating SERPINA3 on cognitive function and glucose homeostasis at both the cellular and animal levels, are necessary.

Clinical studies have demonstrated that certain sulfonylurea hypoglycaemic agents, in addition to their glucose-lowering effects, may exert neuroprotective effects through multiple mechanisms. A prospective cohort study involving 2142 patients with type 2 diabetes mellitus revealed that the long-term administration of glimepiride was associated with a 31% reduction in the risk of cognitive decline compared with non-users (HR = 0.69, 95% CI: 0.52–0.91; *p* = 0.009) [17]. Researchers have postulated that this protective effect might be attributed to the ability of sulfonylureas to activate ATP-dependent potassium channels, increase neuronal energy metabolism, and ameliorate oxidative stress-induced damage. In conjunction with our molecular docking results, we discovered that glimepiride and gliclazide exhibited high binding affinities for the SERPINA3 protein (binding energies of −6.8 and −6.6 kcal/mol, respectively), providing a structural basis for elucidating the neuroprotective mechanisms of sulfonylurea agents. These findings suggest that selecting sulfonylurea hypoglycaemic agents with neuroprotective potential may yield superior clinical benefits in the prevention and treatment of DCI.

Our research, through data analysis and Western blot validation, confirmed that SERPINA3 is a key gene whose expression is significantly upregulated in DCI patients, which aligns with previously reported SERPINA3 upregulation patterns in neurodegenerative disorders and various inflammatory diseases. On the basis of functional enrichment and machine learning screening results, SERPINA3 plays crucial roles in cell cycle regulation and immune–inflammatory pathways, with its elevated expression representing both a defensive mechanism against exogenous stress and potentially reflecting the host’s response to microenvironmental alterations. We examined SERPINA3 expression in DCI patient plasma and conducted protein structure and molecular docking studies, revealing interactions between the alpha-1-antichymotrypsin protein (encoded by SERPINA3) and the sulfonylurea hypoglycaemic agents gliclazide and glepiride. These findings, which are consistent with those of previous cohort studies and clinical controlled trials, provide novel research directions for further investigations into the role of sulfonylurea agents in reducing or delaying the onset of chronic complications in patients with T2DM.

From a translational perspective, SERPINA3, as a potential target within the serine protease inhibitor family, has several advantages. First, SERPINA3 has dual anti-inflammatory and neuroprotective effects. The inhibition of cathepsin G attenuates inflammatory cascade reactions while simultaneously playing a protective role in neurons and oligodendrocytes. This dual mechanism potentially enables concurrent improvement in metabolic parameters and the preservation of neurological function. Second, SERPINA3 is strongly associated with insulin resistance and vascular disorders: SERPINA3 influences multiple cellular signalling axes, including crucial components related to insulin signalling pathways and cardiovascular remodelling. These findings suggest that further optimisation of this molecule or an interruption of its aberrant signalling activities might enhance the comprehensive management of diabetic complications. Furthermore, this gene can interact structurally with existing hypoglycaemic agents. Our molecular docking results indicate that small-molecule drugs can bind to crucial functional domains of SERPINA3, thereby modulating its interactions with downstream pathways. This approach not only holds promise for developing novel therapeutic categories but also provides opportunities for drug repurposing—for instance, through rational chemical modification of existing marketed hypoglycaemic or lipid-lowering agents to increase their binding affinity for SERPINA3 domains, potentially augmenting neuroprotective and anti-inflammatory properties while maintaining glucose-lowering efficacy.

## 4. Materials and Methods

### 4.1. Data Sources

The data for this study were obtained from the publicly available gene expression database GEO (Gene Expression Omnibus, https://www.ncbi.nlm.nih.gov/geo/ accessed on 11 November 2024). The DCI dataset was selected by filtering the datasets containing cortical neuronal samples from normal controls and T2DM patients. There are 11 samples in the GSE161355 dataset, including 5 normal samples and 6 self-reported cortical neuronal samples from patients with T2DM, which were used as the training set. We selected 11 patients in the GSE84422 dataset who were identified as having AD and 11 samples from 11 temporal lobe brain regions of age- and ethnicity- and sex-matched normal controls, which served as the validation set.

### 4.2. Source of Biological Samples

The investigators collected eight blood samples from a large general hospital in Beijing from May–June 2024, four of which were clinically diagnosed patients with T2DM combined with new-onset cognitive impairment (DCI group), and four were sex- and age-matched healthy controls (Con group). All samples were collected with the approval of the Ethics Committee of Peking University (approval number: 2022016) and in strict compliance with the Declaration of Helsinki. All participants signed an informed consent form.

### 4.3. Experimental Design

This study adopted a multistep experimental design, combining bioinformatics analysis and experimental validation to systematically screen and validate the key genes of DCI. The research process included the following parts: an analysis of differentially expressed genes (DEGs); GO (Gene Ontology)/KEGG (Kyoto Encyclopedia of Genes and Genomes) analysis; machine learning screening of key genes (LASSO and SVM-RFE); experimental validation (western blot experiments); protein structure visualisation; binding site detection; and protein–molecule docking.

#### 4.3.1. Analysis of DEGs

This analysis was performed via R software. The ‘limma’ package of R software was used to analyse the differences in the GSE125364 dataset. The DEGs were screened by |logFC| ≥ 0.585 and corrected *p* value < 0.05, and heatmaps and volcano maps were generated with ‘pheatmap’ and ‘ggplot2′.

#### 4.3.2. GO and KEGG Analyses

This analysis was performed via R software (V4.3.3). The R packages ‘clusterProfiler’, ‘org.Hs.eg.db’, and ‘enrichplot’ were used to analyse samples from the GSE125364 dataset. The samples from the GSE125364 dataset were analysed with a *p* value filter of 0.05 and a corrected *p* value filter of 0.05.

#### 4.3.3. Machine Learning Methods for Screening Diagnostic Genes and External Data Validation

The DEGs obtained from the analysis in Section 4.3.1 were screened via LASSO regression and SVM-RFE algorithms. The machine learning approach was implemented as follows: Sample size: 11 samples (5 normal controls and 6 T2DM patients with cognitive impairment). Features (X): 106 differentially expressed genes identified from the differential expression analysis. Target variable (Y): The binary classification of cognitive status (0 for normal controls, 1 for DCI patients). Feature selection process:(a)LASSO regression:

Input matrix: 106 × 11 gene expression matrix;

Optimization criterion: Minimizing mean squared error with L1 regularization;

Cross-validation: 5-fold cross-validation for optimal lambda selection.

(b)SVM-RFE:

Kernel function: Linear kernel;

Ranking criterion: The weight magnitude of SVM classifier;

Feature elimination: The recursive elimination of lowest-ranked features;

Cross-validation: 5-fold cross-validation for performance evaluation.

The final candidate genes were selected by taking the intersection of genes identified by both methods to ensure robustness of selection. The intersection of the genes screened in the two parts was taken to evaluate the diagnostic value of the screened genes through the ROC curve, and the AUC value was calculated. In this part, the R packages ‘glmnet’, ‘e1071’, ‘kernlab’, ‘caret’, and ‘pcaret’ were used as well as ‘pROC’ packages.

### 4.4. Western Blot (WB) Experiment

SERPINA3 (serpin family A member 3), also known as alpha-1-antichymotrypsin, is a plasma protein, so human plasma samples were used in this study for WB experiments.

#### 4.4.1. Plasma Protein Extraction

Plasma proteins were extracted via a plasma protein extraction kit provided by Solarbio (Cat# EX1170; Solarbio, Beijing, China). The procedure was as follows: for every 200 µL of protein extract, 2 µL of protease inhibitor mixture was added, mixed, and incubated on ice. Whole blood samples were centrifuged at 4 °C and 1500× *g* for 10 min, and the cell precipitate was discarded. Each 100 µL plasma sample was mixed with 100–200 µL of protein extract, incubated at 4 °C for 5 min, and then centrifuged at 4 °C and 14,000× *g* for 10 min. The supernatant was pipetted into another clean, precooled centrifuge tube to obtain plasma proteins. The protein concentration was determined via a BCA protein assay kit (Cat# PC0020, Solarbio) according to the manufacturer’s instructions. Equal amounts of total protein were electrophoresed and transferred to a PVDF membrane. After the membranes were blocked and incubated with primary antibodies, they were incubated with the corresponding secondary antibodies. The membranes were then washed and visualised via an enhanced chemiluminescence kit, and the integrated optical density of the protein bands was analysed via ImageJ software (v.1.8.0).

#### 4.4.2. Protein Concentration Measurement and WB Experiment

The protein concentration was first determined via a BCA protein concentration assay kit (Cat# PC0020, Solarbio). Aliquots of protein samples were subjected to SDS–PAGE and transferred to PVDF membranes. The membrane was blocked with 5% skim milk powder for 1 h. The membrane was incubated overnight at 4 °C (for more than 18 h) with a 1:1000 dilution of SERPINA3 antibody (CSB-PA021059ESR1HU, CUSABIO, Wuhan, China). After incubation, the membrane was washed with Tris-buffered saline with Tween buffer (TBST) on a shaker. After incubation, the sample was washed with Tris-buffered saline with Tween buffer (TBST) on a shaker 3 times (5 min each time). After washing, goat anti-human IgG (CSB-PA160506, CUSABIO) was added at a dilution of 1:3000, and the sample was incubated for 2 h at ambient temperature. After incubation, the sample was washed with TBST 3 times (5 min each time) and visualised via an enhanced chemiluminescence kit (SR5580, HARVEYBIO, Beijing, China), and the optical density of the protein bands was analysed via ImageJ software.

### 4.5. SERPINA3 Protein Structure Analysis and Molecular Docking

The structure of SERPINA3 and its binding site characteristics were comprehensively analysed via PyMOL software. First, the SERPINA3-encoded alpha-1-antichymotrypsin protein structure (UniProt ID: P01011, resolution, 2.1 Å) was downloaded and loaded from the AlphaFold database. The overall secondary structure of the protein was clearly demonstrated by setting the display mode (hide other elements and show only the ‘cartoon’ backbone) and background colour (white). In addition, to highlight the functional regions, the following specific settings were used: the active site region (358–374, i.e., the reaction centre loop RCL region) was selected as the ‘active_site’, displayed as sticks, and coloured in orange to enhance the recognition of the region. For further analysis, potential binding pockets were identified by surface modelling, the ‘show surface’ command was used to generate molecular surfaces, and ‘vacuum_electrostatics’ was turned on to calculate the electrostatic potential field and obtain the electrostatic distribution information of the binding sites. The ‘show surface’ command was used to generate the molecular surface, and ‘vacuum_electrostatics’ was turned on to calculate the electrostatic potential field to obtain the electrostatic distribution of the binding site. Moreover, the PyMOL (V3.12) selection command was used to screen potential ligand binding residues within 8 Å of the active site, highlighting hydrophobic residues (e.g., Met358, Pro357) and polar/charged residues (e.g., Ser359, Arg217). In addition, external inserts (POCASA) were used to assess the volume and surface area of the pockets and the geometrical properties of the binding regions.

To assess the binding energies and interaction patterns between several screened hypoglycaemic agents and their targets, we used AutodockVina 1.2.2 (http://autodock.scripps.edu/, accessed on 11 November 2024) software. The molecular structures of five clinically used hypoglycaemic drugs (metformin, gliclazide, glepiride, repaglinide, and liraglutide) were obtained from the PubChem compound database (https://pubchem.ncbi.nlm.nih.gov/, accessed on 11 November 2024). We first prepared the protein and ligand files by converting all protein and molecular files to PDBQT format, removing all water molecules and adding polar hydrogen atoms. Grid boxes were centred to cover the structural domains of each protein and to accommodate free molecular motion. The docking pocket was set up as a 40 Å × 40 Å × 40 Å square pocket with a grid point distance of 0.05 nm, and molecular docking was performed via PyMOL software.

## 5. Conclusions

In summary, SERPINA3/Serpina3c, a serine (or cysteine) peptidase inhibitor, has demonstrated multifaceted regulatory roles in non-neoplastic diseases, including neurodegenerative disorders, cardiovascular diseases, inflammatory conditions, and metabolic syndrome. Our research has established the pivotal role of SERPINA3 in diabetes-related cognitive impairment, suggesting that the modulation of this gene or its associated signalling pathways may provide novel approaches for DCI prevention, diagnosis, and treatment. Future research directions should focus on the following: first, validating the causal relationship between SERPINA3 and DCI through large-scale clinical cohorts; second, elucidating the precise regulatory mechanisms of SERPINA3 through genetic engineering approaches; and third, developing innovative therapeutic strategies that combine glucose-lowering and neuroprotective properties through structural upgrading and the functional integration of existing hypoglycaemic or anti-inflammatory agents. With ongoing advances in molecular biology and drug development technologies, SERPINA3-specific interventions are expected to broaden the horizons for the comprehensive management of diabetes and its related complications.

## Figures and Tables

**Figure 1 ijms-26-01947-f001:**
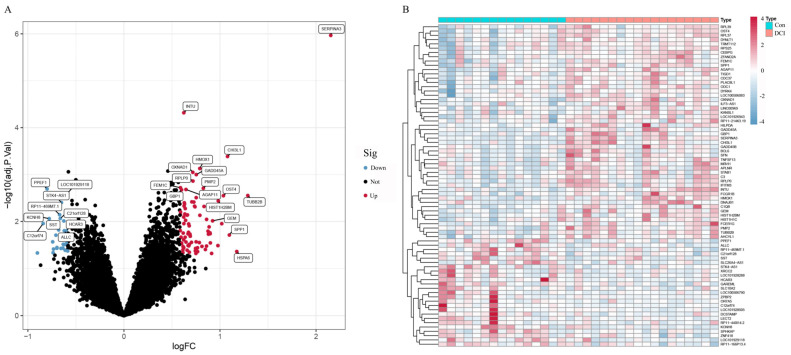
Analysis of differentially expressed genes in DCI patients and the control group. (**A**) Volcano plot showing the distribution of DEGs (red indicates upregulated genes, blue indicates downregulated genes, and black indicates genes with no significant difference). (**B**) Heatmap showing the expression patterns of DEGs in DCI and control samples (red indicates high expression and blue indicates low expression).

**Figure 2 ijms-26-01947-f002:**
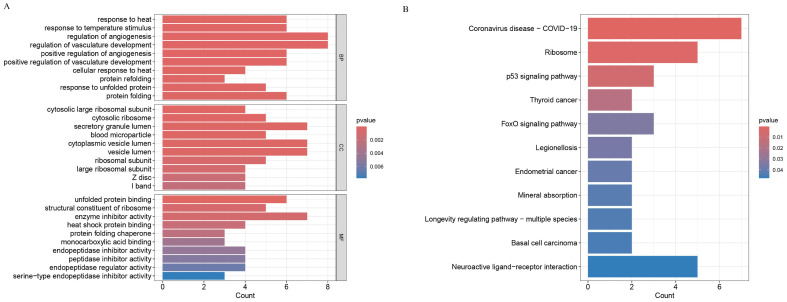
Functional enrichment analysis of differentially expressed genes. (**A**) Results of the GO enrichment analysis (including biological process BP, cellular component CC, and molecular function MF). (**B**) The results of the KEGG pathway enrichment analysis (eleven signalling pathways significantly enriched).

**Figure 3 ijms-26-01947-f003:**
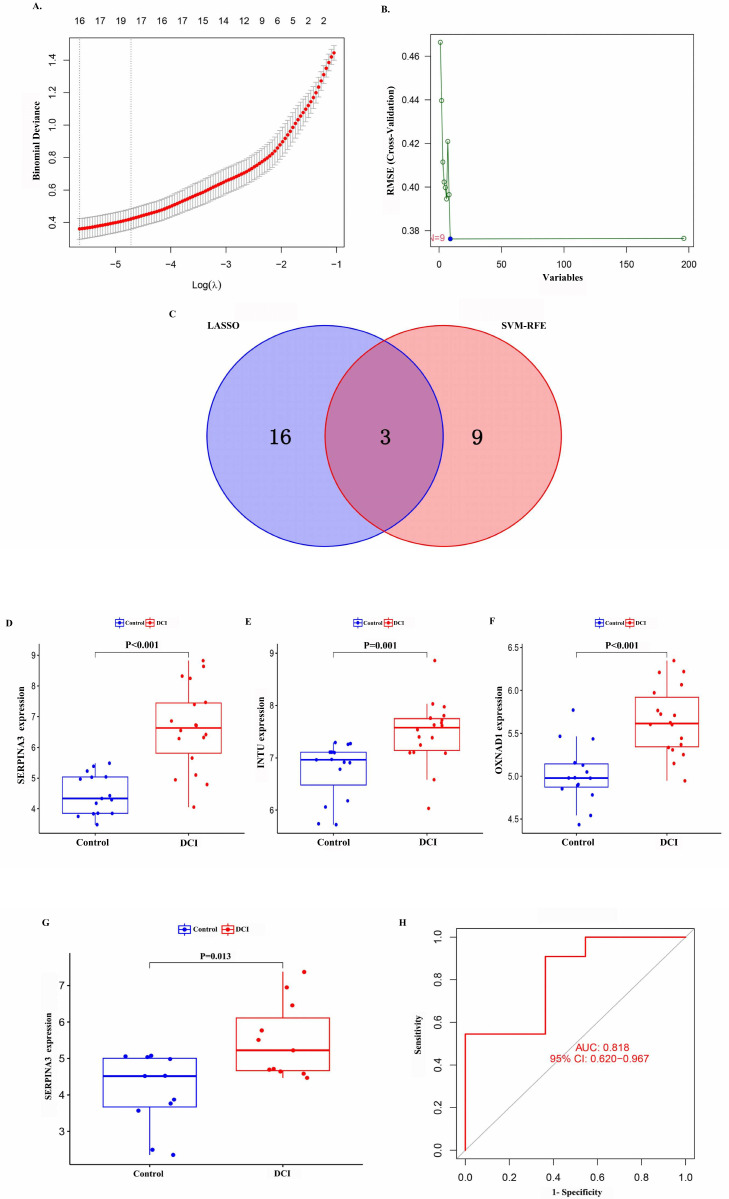
Machine learning screening of diagnostic genes and validation results. (**A**) Results of LASSO regression analysis. (**B**) The results of SVM-RFE feature selection. (**C**) Venn diagrams of genes screened by the two methods. (**D**–**F**) Differences in the expression of SERPINA3, INTU, and OXNAD1 in the training set. (**G**) Validation of the expression of SERPINA3 in the validation set. (**H**) Analysis of the ROC curve of SERPINA3.

**Figure 4 ijms-26-01947-f004:**
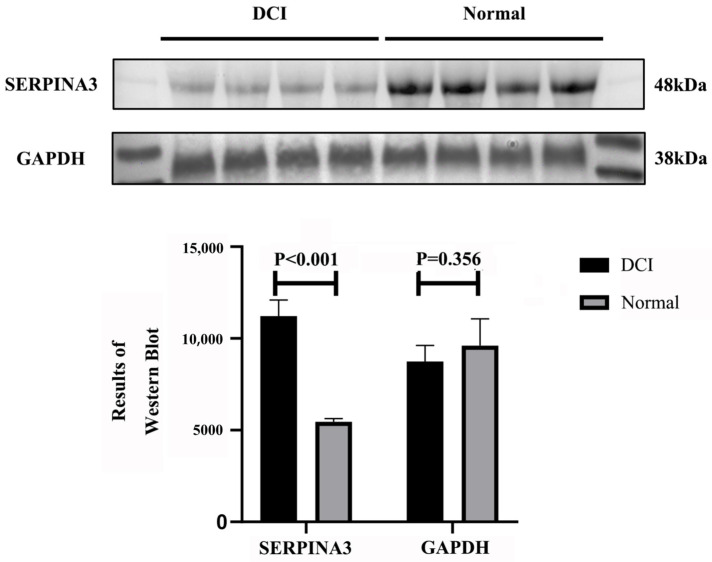
Results of the WB validation experiments for SERPINA3 protein expression. Western blot results of plasma samples from the DCI group and control group.

**Figure 5 ijms-26-01947-f005:**
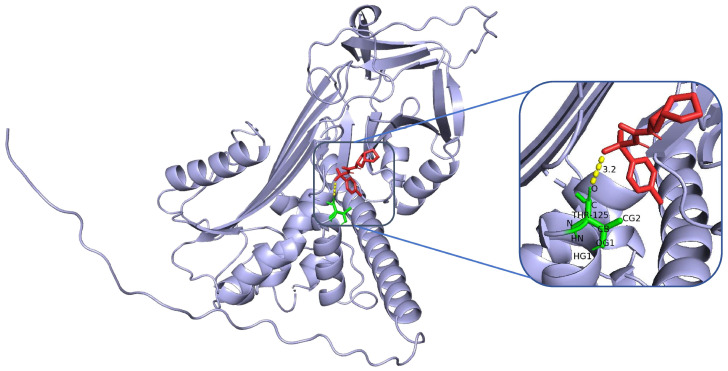
Molecular docking analysis of gliclazide with the SERPINA3 protein.

**Figure 6 ijms-26-01947-f006:**
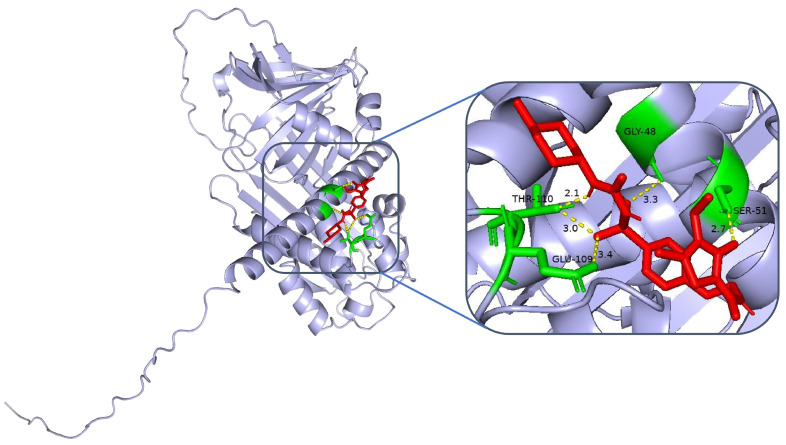
Molecular docking analysis of glimpuridine with the SERPINA3 protein.

## Data Availability

The data supporting the findings of this study are openly available in the GENE EXPRESSION OMNIBUS (GEO) at https://www.ncbi.nlm.nih.gov/geo/ (accessed on 4 May 2024).

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
