# Peer review of "SERPINA3: A Novel Therapeutic Target for Diabetes-Related Cognitive Impairment Identified Through Integrated Machine Learning and Molecular Docking Analysis"

_ijms, 2025, doi:10.3390/ijms26051947_

Round 1

Reviewer 1 Report

Comments and Suggestions for Authors

The article presents a solid methodology in the search for the correlation between cognitive impairment caused by type 2 diabetes and Serpina 3.

The results are consistent with the methodology used.

The protective mechanism of sulfonylureas with respect to the action of Serpina 3 and therefore preventing cognitive damage is widely discussed.

Comments and Suggestions for Authors: To analyze whether Serpina 3 may be a potential target for other hypoglycemic agents such as semaglutide.

Author Response

Comments:

The article presents a solid methodology in the search for the correlation between cognitive impairment caused by type 2 diabetes and Serpina 3.

The results are consistent with the methodology used.

The protective mechanism of sulfonylureas with respect to the action of Serpina 3 and therefore preventing cognitive damage is widely discussed.

Comments and Suggestions for Authors: To analyze whether Serpina 3 may be a potential target for other hypoglycemic agents such as semaglutide.

Response:

Thank you for your thoughtful suggestion regarding the potential analysis of semaglutide. We greatly appreciate your insight into expanding the scope of our investigation. However, we would like to respectfully explain our methodological considerations that led to the exclusion of semaglutide from our molecular docking analyses.

Our structural analysis of SERPINA3, conducted using PyMOL software, revealed specific binding site characteristics that guided our molecular selection criteria. We identified three distinct binding regions:

The primary binding pocket, located in the RCL region, features a hydrophobic core surrounding polar residues (notably Ser359). This pocket's architecture is particularly suited for small-molecule binding.

Secondary binding pockets situated between β-sheet layers, characterised by larger hydrophobic cavities.

Regulatory sites positioned distal to the active site, comprised of charged residue clusters (including Arg217, Glu354, and Lys241).

In pursuit of methodological efficiency and precision, we employed a short-term optimisation approach. Our molecular selection criteria were rigorously based on:

  1. a) The geometric and physicochemical properties of the binding sites, particularly charge distribution and hydrophobicity
  2. b) Structural similarity to established protease inhibitors (Camostat and Nafamostat)
  3. c) Compatibility with the identified binding pocket characteristics

Semaglutide, whilst an effective GLP-1 receptor agonist, possesses a molecular structure that falls outside our established selection parameters. As a large peptide molecule, its structural properties differ significantly from the small-molecule paradigm we established for this study. Therefore, proceeding with molecular docking analysis of semaglutide would not align with our methodological framework and could potentially introduce inconsistencies in our systematic approach.

We believe this focused methodology has allowed us to maintain scientific rigour and generate more reliable and comparable results within our defined scope of investigation.

We trust this explanation clarifies our methodological choices and welcome any further discussion on this matter.

Reviewer 2 Report

Comments and Suggestions for Authors

The authors present a study on the discovery of a novel target protein for diabetes-related cognitive impairment using machine learning and molecular docking. The manuscript is well-structured and in good shape. However, I have one comment regarding the machine learning part.

The description of the machine learning approach is somewhat unclear. It looks like that the authors used two feature selection methods, SVM-RFE and LASSO. However, it is unclear what features (X) were used and what the target variable (Y) was, as well as the dataset size, etc. Please clarify this in the Methods section to enhance transparency and reproducibility.

Author Response

Comments:

The authors present a study on the discovery of a novel target protein for diabetes-related cognitive impairment using machine learning and molecular docking. The manuscript is well-structured and in good shape. However, I have one comment regarding the machine learning part.

The description of the machine learning approach is somewhat unclear. It looks like that the authors used two feature selection methods, SVM-RFE and LASSO. However, it is unclear what features (X) were used and what the target variable (Y) was, as well as the dataset size, etc. Please clarify this in the Methods section to enhance transparency and reproducibility.

Response:

We are grateful for your constructive feedback regarding the clarity of our machine learning methodology. Your comments have helped us significantly improve the transparency and reproducibility of our work.

In response to your concerns, we have made substantial revisions to the manuscript, particularly in the Methods section (4.3.3), where we have now provided a comprehensive description of our machine learning approach. The modifications include:

  1. Detailed Dataset Characteristics:
  • We have explicitly stated the sample size (11 samples, comprising 5 normal controls and 6 T2DM patients with cognitive impairment)
  • We have clearly defined the features (X) as 106 differentially expressed genes
  • We have specified the target variable (Y) as the binary classification of cognitive status
  1. Thorough Feature Selection Process:
    For LASSO regression:
  • We have detailed the input matrix dimensions (106 × 11)
  • We have specified the optimization criterion and cross-validation approach
  • We have explained the parameter selection process

For SVM-RFE:

  • We have outlined the kernel function choice
  • We have described the ranking criterion and feature elimination process
  • We have detailed the cross-validation methodology

Furthermore, we have enhanced the Discussion section with a methodological consideration paragraph that addresses the robustness of our approach despite the relatively small sample size, emphasising how the combination of two independent methods (LASSO regression and SVM-RFE) helps mitigate potential bias.

We believe these revisions have substantially improved the manuscript by:

  1. Enhancing the transparency of our machine learning methodology
  2. Ensuring the reproducibility of our analytical approach
  3. Demonstrating the rigour of our methodological framework

We trust these modifications adequately address your concerns and provide the clarity you sought regarding our machine learning approach. We welcome any additional feedback you may have.

Round 2

Reviewer 2 Report

Comments and Suggestions for Authors

Thanks for the detailed response.  my comments has been solved.